# The Effect of Semaglutide on Blood Pressure in Patients without Diabetes: A Systematic Review and Meta-Analysis

**DOI:** 10.3390/jcm12030772

**Published:** 2023-01-18

**Authors:** Cormac Kennedy, Peter Hayes, Sulafa Salama, Martina Hennessy, Federica Fogacci

**Affiliations:** 1Department of Pharmacology and Therapeutics, School of Medicine, Trinity College Dublin, D08W9RT Dublin, Ireland; 2Wellcome-HRB Clinical Research Facility, St James Hospital, D08W9RT Dublin, Ireland; 3Department of Pharmacology, Trinity Health Sciences Centre, St James Hospital, D08W9RT Dublin, Ireland; 4Health Research Institute, University of Limerick, V94 T9PX Limerick, Ireland; 5Hypertension and Atherosclerosis Research Group, Medical and Surgical Sciences Department, Alma Mater Studiorum University of Bologna, 40138 Bologna, Italy

**Keywords:** blood pressure, hypertension, body weight, obesity, glucagon-like peptide-1 receptor agonists, semaglutide, weight loss, randomized controlled trials, STEP trials

## Abstract

(1) Background: Recent advances in the pharmacological treatment of obesity with glucagon-like peptide-1 receptor agonists (GLP-1 RA) highlight the potential to target excess body weight to improve blood pressure (BP). This review aimed to determine the BP reduction in trials of semaglutide for weight reduction in patients without diabetes. (2) Methods: Relevant studies were identified via a search of research databases. Studies were screened to include randomized controlled trials (RCTs) of semaglutide versus a placebo in adults. Pooled and sensitivity analyses were performed, and risk of bias was assessed. (3) Results: six RCTs, with 4744 participants, were included in the final analysis. At baseline, the cohorts in these studies had a mean BP in the normotensive range. The mean difference in systolic BP was −4.83 mmHg (95% CI: −5.65 to −4.02), while that for diastolic BP was −2.45 mmHg (95% CI: −3.65 to −1.24). All included studies were of a high methodological quality. (4) Conclusions: A clinically significant reduction in BP was evident following semaglutide treatment in normotensive populations without diabetes. The effect of semaglutide in those with obesity and hypertension is as yet undetermined. Targeting excess body weight may be a novel therapeutic strategy for these patients.

## 1. Introduction

Globally, the prevalence of obesity and hypertension is increasing rapidly [1,2]. It is estimated that 60% to 70% of the hypertension burden in adults is due to adiposity [3]. Across numerous cohorts of patients with hypertension, approximately half are also obese [4,5]. Both conditions are associated with cardiovascular disease (CVD) and involve numerous interacting pathways. These interactions include renal salt handling, increased renin–angiotensin–aldosterone system activity, and altered peripheral vascular resistance [6,7]. Underlying these mechanisms is an inflammatory milieu associated with adipose tissue that increases oxidative stress and advances vascular aging [8].

Weight reduction is achievable by numerous means. Studies targeting weight loss by dietary interventions consistently demonstrate an associated reduction in blood pressure (BP) [9,10]. However, weight loss is rarely maintained. A review of the effect of medical weight reduction interventions on BP found a modest effect may be present but evidence was limited [11]. Evidence from one meta-analysis of patients post bariatric surgery showed that 75% of those with hypertension achieved systolic BP (SBP) in the normal range and had on average a reduction of 4.2 mmHg SBP, following the surgery [11].

Interestingly, studies may not capture the totality of the BP reduction, as patients are frequently reported to reduce their anti-hypertensive treatment when losing weight, but this outcome may not be reported or quantified.

In 2021, reports from the Semaglutide Treatment Effect for People with Obesity (STEP) trials provided evidence of the effect of semaglutide, a glucagon-like peptide-1 receptor agonist (GLP-1 RA), on reducing body weight in patients without diabetes [12,13,14]. An initial review of the trial results suggests a positive effect on BP. The aim of this review was to systematically assess the effect of semaglutide on BP in participants with obesity but without diabetes in the setting of a randomized controlled trial (RCT), as well as to capture the effect of semaglutide on the number of anti-hypertensive medications that patients were prescribed during the trial period. This work intends to inform a weight-centric approach to control BP with GLP-1 RAs such as semaglutide.

## 2. Materials and Methods

The study was designed in agreement with the 2020 Preferred Reporting Items for Systematic Reviews and Meta-Analysis (PRISMA) statement guidelines and it was registered in the PROSPERO database (Registration number CRD42022350115) [15]. Due to study design, neither Institutional Review Board (IRB) approval nor patient informed consent were required.

EMBASE, Medline with Ovid, CINAHL, The Cochrane Library, Google Scholar and Web of Science by Clarivate databases were searched, with no language restriction, using the following keywords and MeSH terms: semaglutide, ozempic, Wegovy, Rybelsus, blood pressure, blood pressure determination, hypertension, blood pressure monitoring, clinical trial, randomized clinical trial, random allocation, placebo. The reference list of identified papers was manually checked for additional relevant articles. The literature was searched from database inception to 1 July 2022.

All paper abstracts were screened in Covidence^®^ (Covidence systematic review software version 2.0, Veritas Health Innovation, Melbourne, Australia) by two reviewers (CK, PH) in an initial process to remove ineligible articles. The remaining articles were obtained in full-text and assessed again by the same two researchers, who evaluated each article independently and carried out data extraction and quality assessment. Disagreements were resolved by discussion with a third party (SS).

Original studies were included if they met the following criteria: (i) being a clinical trial with either multicentre or single-centre design, (ii) having an appropriate placebo-controlled design for semaglutide treatment, (iii) investigating the effect of semaglutide where BP was a reported outcome, (iv) enrolling human adults who were without a diagnosis of diabetes mellitus, (v) having N of at least 20 per group, (vi) having a minimum of 12-week follow-up. Studies were excluded if they contained overlapping subjects with other studies.

Data extraction and database typing were reviewed by the principal investigator before the final analysis, and doubts were resolved by mutual agreement among the authors.

A systematic assessment of risk of bias (RoB) in the included studies was performed using the Cochrane criteria [16]. RoB assessment was performed independently by 3 reviewers (CK, PH, SS); disagreements were resolved by a consensus-based discussion. The following items were used: adequacy of sequence generation, allocation concealment, blinding addressing of dropouts (incomplete outcome data), selective outcome reporting, and other probable sources of bias.

Meta-analysis was conducted using Comprehensive Meta-Analysis (version 3.0, Biostat, Englewood, NJ, USA, 2022). Net changes in the investigated parameters were calculated by subtracting the BP at baseline from the post-intervention BP, in the active-treated group and control groups. This provided the mean change from baseline. If the outcome measures were reported as mean and 95% confidence interval (CI) or standard errors of the mean, standard deviations were estimated using the methods described by Higgins et al. [17]. To avoid a double-counting problem, in trials comparing multiple treatment arms versus a single control group, the number of subjects in the control group was divided by the required comparisons. The findings from studies were combined using a random-effects model (using the DerSimonian–Laird method) and the generic inverse variance method. Heterogeneity was quantitatively assessed using the Higgins index (I2). Effect sizes for blood pressure were expressed as mean differences (MD) and 95% confidence interval (CI). To evaluate the influence of each study on the overall effect size, sensitivity analysis was conducted using the leave-one-out method. Two-sided *p*-values ≤ 0.05 were considered as statistically significant for all tests.

A sensitivity analysis was conducted by omitted the phase II dose finding study so that only phase III studies were analysed. This analysis was performed to assess the likely treatment effect with the 2.4 mg dose of semaglutide, which is the licensed weight loss dose. A leave-one-out analysis was also performed for treatment effect on SBP and DBP. As a component of the sensitivity analyses, fixed-effects models were used to combine the data from the studies.

Further meta-regression analysis was conducted to determine the influence of study characteristics on the treatment effect (SBP). The characteristics of interest were duration of follow-up and dose of semaglutide.

Potential publication biases were explored using visual inspection of Begg’s funnel plot asymmetry, Begg’s rank correlation test, and Egger’s weighted regression test. Two-sided *p*-values ≤ 0.05 were considered statistically significant.

## 3. Results

### 3.1. Flow and Characteristics of the Included Studies

Six hundred and eleven potential studies were identified by our initial search. We extracted data from six studies (see Prisma diagram, Figure 1).

The six studies randomized 4744 participants to intervention and control arms. Most were female and white. Full details of these six studies can be found in Table 1.

The studies were conducted across multiple sites and countries. The eligibility criteria for trial entry and the selected trial outcomes were consistent across all included studies. The intervention in the five phase III RCTs (STEP Trials 1,3,4,5,8) was semaglutide at a dose of 2.4 mg. A much lower dose was used in the earliest study from 2018 (O’Neill). The follow-up period ranged from 52 to 104 weeks. Of note, the trial design for STEP 4 required an active run-in phase followed by a withdrawal of semaglutide treatment for the placebo arm. Therefore, the change in BP was inverted as the BP of the placebo group increased after randomization.

### 3.2. Meta-Analysis Results

The mean difference in SBP, seen in Figure 2, was −4.83 mmHg (95% CI: −5.65 to −4.02). The studies had low heterogeneity (I2 = 0%).

The leave-one-out analysis demonstrated the effect on SBP was robust and not driven by a single study, with mean differences in SBP for individual studies varying from −4.63 to −5.04 mmHg (Appendix A). The mean difference in DBP was −2.45 mmHg (95% CI: −3.65 to −1.24) (Figure 3). The heterogeneity was high for this analysis (I2 = 87.5%). The treatment effect in individual studies varied from −0.70 to −4.90 mmHg. The leave-one-out analysis demonstrated the effect on DBP was not driven by a single study.

The sensitivity analysis seen in Figure 4, including the phase III studies only, provided a similar result with a mean difference in SBP of −4.83 mmHg (95% CI: −5.72 to −3.94).

The fixed-effects model for SBP change showed an unchanged effect (−4.83 mmHg, 95% CI: −5.65 to −4.02). The analysis of change in DBP using a fixed-effects model resulted in less of an effect, −1.57 mmHg (95%CI: −1.75 to −1.39) versus −2.45 mmHg for the random effects model. It is evident that STEP 1 dominates the analysis due to its small variance (Appendix A).

Meta-regression analysis did not suggest a time-dependent effect on SBP (slope: −0.0089; 95%CI: −0.0690 to 0.0513; Z-value = −0.29; two-tailed *p* > 0.05) or DBP (slope: −0.0511; 95%CI: −0.1073 to 0.0052; Z-value = −1.78; two-tailed *p* > 0.05). In addition, no dose-dependent effect on SBP (slope: −0.0194; 95%CI: −1.0546 to 1.0158; Z-value = −0.04; two-tailed *p* > 0.05) or DBP (slope: −0.4464; 95%CI: −1.3603 to 0.4674; Z-value = −0.96; two-tailed *p* > 0.054) was apparent.

Only two studies (STEP 1 and 4) reported a change in anti-hypertensive medications. In STEP 1, 34.3% of those on anti-hypertensive medications either decreased or stopped them; this was 16% in the placebo group. The equivalent figures were 25.5% and 11.9% for STEP 4.

### 3.3. Risk of Bias Assessment

All trials were of high quality with clear randomization sequence generation, quadruple blinding and extensive data reporting. Allocation concealment was appropriate, though a bias was possibly introduced when those on placebo did not lose weight. However, the inclusion of a behavioural (diet and physical activity) intervention may have mitigated this risk, and drop-out rates were similar across arms in the included studies. Selective reporting was limited by prespecified trial procedures in registered protocols. All studies were funded by the manufacturer and authorization holder of the drug.

### 3.4. Publication Bias

Visual inspection of Begg’s funnel plot (Appendix A) revealed a slight asymmetry, suggesting potential publication bias for the effect of semagutide on SBP. This asymmetry was imputed to three potentially missing studies on the left-side of the funnel plot, which increased the estimated effect size to −5.28 (95%CI: −5.99 to −4.57). However, Egger’s linear regression and Begg’s rank correlation did not confirm the presence of publication bias (*p* > 0.05 always). The classic fail-safe N test suggested that 303 studies with negative results would be needed to bring the estimated effect size on SBP to a nonsignificant level (*p* > 0.05).

## 4. Discussion

This systematic review provides an estimate of the pooled effect of semaglutide on BP in patients with obesity but without diabetes. Of note, the mean BP of cohorts in the included studies was in the normal range at baseline. Six studies were included, and the treatment effect was approximately a 5 mmHg reduction in systolic BP and a 2.5 mmHg reduction in diastolic BP. The included studies were of high methodological quality with a low risk of bias. The effect size was not dominated by one study, as evident from the leave-one-out analysis. Variation in study design in terms of semaglutide dose or duration of follow-up did not significantly alter the estimated treatment effect. When the analysis included large studies at the licensed weight-loss dose of semaglutide (2.4 mg), the treatment effect remained approximately 5 mmHg. The effect of semaglutide on the anti-hypertensive medication burden suggests that trial participants were prescribed less medication following semaglutide treatment. As this outcome was reported in two studies only, this finding was not amenable to a pooled analysis.

This study brings into sharp focus the possibility of a new paradigm for the treatment of hypertension in patients with obesity [21]. A weight-centric approach to hypertension treatment must now be carefully considered. A population of patients who might benefit most in terms of BP reduction, while also mitigating their cardiovascular (CV) risk, are patients with obesity and resistant hypertension (RH) or patients with difficult-to-control blood pressure who may not meet the criteria for RH. Targeting excess body weight in these patients with an agent now proven to reduce weight and BP in a normotensive population appears eminently sensible, if yet unproven.

The evidence provided by this study is consistent with that of previous studies of GLP-1 Ras as well as other weight loss interventions. In diabetic populations, glucagon-like peptide-1 receptor agonist (GLP-1 RAs) and sodium–glucose co-transporter-2 inhibitors have been associated with reductions in blood pressure [22]. Similar effects might be expected in those without diabetes, particularly if weight reduction is a central component of this effect. However, until now, there has been no dedicated analysis to determine the effect of newer GLP-1 RAs such as semaglutide on BP. From this work, it is evident that treatment with semaglutide provides a clinically meaningful reduction in BP. This finding is consistent with the findings of previous studies.

A systematic review including eight clinical trials and 9424 participants undergoing treatment with glucagon-like peptide-1 receptor agonists (GLP-1 RAs), such as semaglutide and liraglutide, with obesity but without diabetes resulted in a mean difference of −4.4 mmHg in SBP and a 7.1 kg decrease in body weight [23]. A further systematic review including three trials and 3375 participants demonstrated a mean difference of −4.62 mmHg in SBP and body weight reduction of 11.9 kg with weekly subcutaneous semaglutide [24]. The body weight reduction across the STEP trials in this analysis varied from 10.6 kg to 13.4 kg with the approximate mean baseline body weight of 105 kg (104.5 kg for STEP 8 to 107.2 kg for STEP 4). It is uncertain whether the reduction in BP is directly related to the body weight reduction, though this is likely a central factor.

Previous work has examined the effect of weight reduction on blood pressure. However, methods of weight reduction that might best result in BP reduction are unclear, in terms of both the magnitude of the effect and its sustainability. A recently updated Cochrane review of weight loss diets in populations with hypertension identified eight trials with 2100 participants on which to base their analysis [25]. They concluded that these diets were associated with a reduction in BP; a mean difference of −4.5 mmHg in SBP was reported based on three trials with an adequately reported mean difference in SBP. The associated weight reduction was approximately 4 kg. Importantly, the authors noted that two of the included trials used withdrawal of anti-hypertensive medications as a primary outcome.

A meta-analysis of randomized controlled trials (RCTs) involving nonpharmacological interventions to reduce body weight, based on data from twenty-five trials and 4874 participants, reported that each kilogram of weight reduction may result in at least 1 mmHg reduction in BP [9]. The mean reduction in SBP was −4.4 mmHg, with a mean reduction in weight of 5.1 kg. Interestingly, this same meta-analysis suggested a greater SBP reduction in those on anti-hypertensive medication compared with untreated populations (7.0 mmHg versus 3.8 mmHg). As well as the weight reduction, a factor contributing to lower BP may be the reduction in salt intake, which is likely to result from dietary approaches to weight reduction. The effect of reduced salt intake on BP has been established by the Dietary Alterations to Stop Hypertension Study [26].

The mechanisms by which GLP1-RAs reduce cardiovascular disease in diabetic populations are subject to much discourse. Mediation analysis suggests that the effect is greater than that expected from the reduction in the factors of cardiovascular disease such as BP [27]. When a medication analysis examining the effect of liraglutide on CVD was performed, a similar modest contributory effect of BP reduction was reported [28]. Cardiovascular outcome trials of GLP1-RAs in populations without diabetes are ongoing but have yet to report. Therefore, a mediation analysis of their effect on CVD is not yet possible. However, a mediation analysis of contributory factors to the reduction in BP should be the next step to further elucidate the evident effect of semaglutide on BP in patients without diabetes. This will identify the contribution of weight loss to the reduction in BP as well as other potential mediators such as natriuresis, which has been previously described [29], and reductions in inflammation.

A strength of the studies included in this analysis is that a background behavioural intervention, which involved both dietary changes and increased physical activity, was incorporated into all trials for both intervention and control arms. In most trials, these interventions also resulted in a modest 1.0 to 1.6 mmHg reduction in SBP in control groups. However, this must be considered in the context of the typically normotensive participants recruited for these trials. Another strength is the homogeneity of the trials in terms of the intervention, the eligibility criteria and the outcome ascertainment. The studies were of high methodological quality, incorporating blinding, placebo control and rigorous follow-up into the trial conduct as well as describing these methods prospectively in registered protocols. Strengths of this analysis include adherence to a prespecified protocol with a clear research question at the outset.

A number of potential issues are apparent from these studies. The populations are predominantly female and white, which may limit the conclusions for other populations. Studies with longer follow-up had smaller sample sizes. Given the questionable sustainability of weight reduction interventions in general, larger and longer follow-up periods are desirable. Moreover, the BP reduction in STEP 8 is larger than that of its counterparts. The small and more comorbid placebo group may have been a factor for this result. However, the leave-one-out analysis did not suggest STEP 8 had a dominant effect on the analysis presented here.

The limitations of this work include the narrow focus of the intervention, which was semaglutide only. Other GLP-1 RAs were not included because many of these agents are older, less effective at weight reduction and therefore not reflective of contemporary treatment of excess body weight. Of interest is the recently published SURMOUNT-1 trial that assessed weight reduction for patients without diabetes with tirzepatide, a dual GLP-1 and GIP receptor agonist [30]. This study reported a SBP reduction of −6.2 (95% CI: −7.7 to −4.8) in a trial population with a normal BP at randomization. In addition, the inclusion of BP measurements at numerous timepoints may have provided a richer analysis. However, this was not feasible as the studies only reported the BP measurement at the study conclusion in sufficient detail for analysis. As the number of included studies was relatively few, the use of meta-regression may be questionable. However, most studies were large RCTs, and the authors here have requested patient-level data for these studies to further analyse the effect of semaglutide on BP. Lastly, an objective of this study was to determine the effect of semaglutide on anti-hypertensive medication burden. However, that was not possible due to the limited publication of this outcome in the available study reports.

## 5. Conclusions

This systematic review focused on the effect of semaglutide treatment on blood pressure in randomized clinical trials recruiting patients without diabetes. There is now high-quality evidence to suggest that semaglutide results in a clinically significant reduction in BP in these trial cohorts, which had a mean baseline BP in the normal range. It is likely that the BP-lowering effect of semaglutide in those who are hypertensive is likely greater than the 4.83 mmHg reported here, but the true effect is yet to be determined. Patients with obesity and difficult-to-control blood pressure may particularly benefit from a weight-centric approach to hypertension treatment.

## Figures and Tables

**Figure 1 jcm-12-00772-f001:**
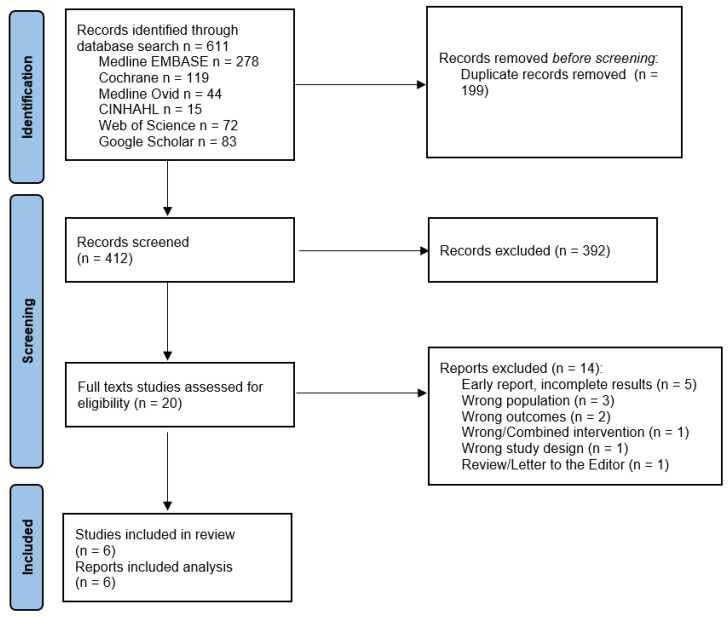
Prisma Diagram [15].

**Figure 2 jcm-12-00772-f002:**
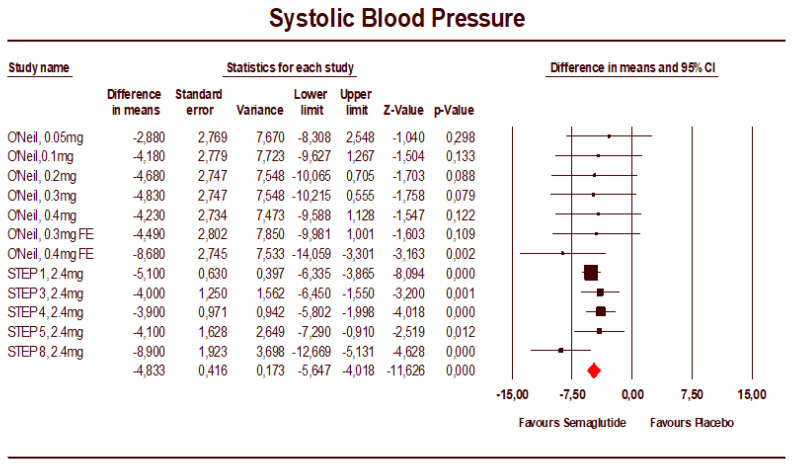
Forest plot of semaglutide vs. placebo showing the pooled weighted mean difference for SBP (random effects model) [18].

**Figure 3 jcm-12-00772-f003:**
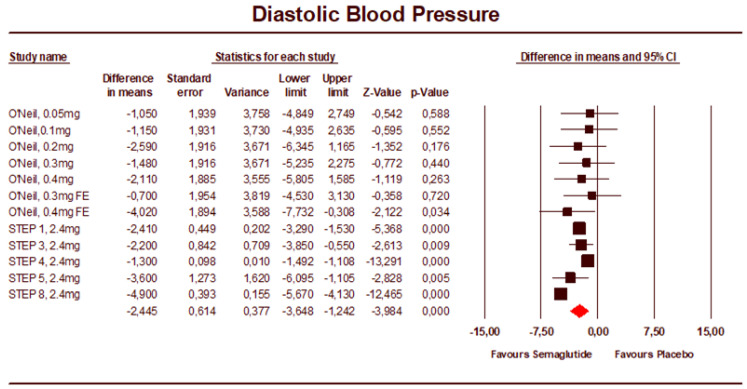
Forest plot of semaglutide vs. placebo showing the pooled weighted mean difference for DBP (random effects model) [18].

**Figure 4 jcm-12-00772-f004:**
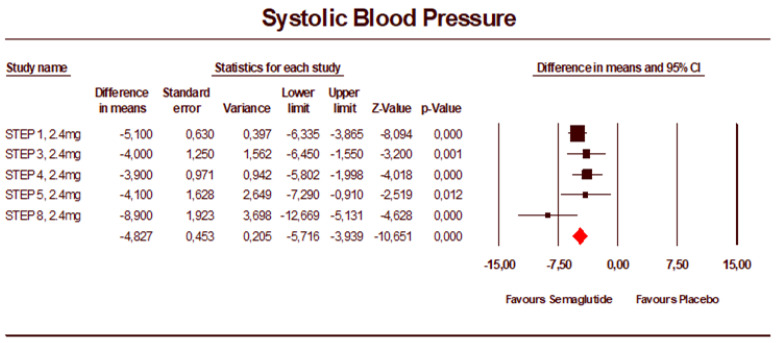
Forest plot of the phase 3 studies of semaglutide 2.4 mg vs. placebo showing the pooled weighted mean difference for SBP (random effects model).

**Table 1 jcm-12-00772-t001:** Main characteristics of the clinical trials included in the systematic review and meta-analysis.

Author, Year	Study	Design	Main Inclusion Criteria	Follow-Up	Study Group	Enrolled Individuals (N)	Men
O’Neil PM, 2018 [18]		Randomized, double-blind, placebo and active-controlled, dose-ranging, phase 2 trial	Adults without diabetes and with a BMI of 30 kg/m^2^ or greater	52 weeks	Semaglutide 0.05 mg	103	36 (35%)
					Semaglutide 0.1 mg	102	36 (35%)
					Semaglutide 0.2 mg	103	37 (36%)
					Semaglutide 0.3 mg	103	37 (36%)
					Semaglutide 0.4 mg	102	36 (35%)
					Semaglutide 0.3 mg FE	102	36 (35%)
					Semaglutide 0.4 mg FE	103	36 (35%)
					Placebo	136	48 (35%)
Wilding JPH, 2021 [12]	STEP-1	Randomized, double-blind, placebo-controlled, phase 3 trial	Adults without diabetes and with a BMI of 30 kg/m^2^ or greater OR a BMI of 27 kg/m^2^ or greater with one or more treated or untreated weight-related coexisting conditions	68 weeks	Semaglutide 2.4 mg	1306	351 (26.9%)
					Placebo	655	157 (24%)
Wadden TA, 2021 [13]	STEP-3	Randomized, double-blind, placebo-controlled, phase 3 trial	Adults without diabetes and with a BMI of 30 kg/m^2^ or greater OR a BMI of 27 kg/m^2^ or greater with at least one comorbidity	68 weeks	Semaglutide 2.4 mg	407	92 (22.6%)
					Placebo	204	24 (11.8%)
Rubino D, 2021 [14]	STEP-4	Randomized, placebo-controlled, phase 3a trial	Adults without diabetes and with a BMI of 30 kg/m^2^ or greater OR a BMI of 27 kg/m^2^ or greater with at least one comorbidity	48 weeks	Semaglutide 2.4 mg	535	106 (19.8%)
					Placebo	368	63 (23.5%)
Garvey WT, 2021 [19]	STEP-5	Randomized, double-blind, placebo-controlled, phase 3 trial	Adults without diabetes and with a BMI of 30 kg/m^2^ or greater OR a BMI of 27 kg/m^2^ or greater with at least one comorbidity	104 weeks	Semaglutide 2.4 mg	152	29 (19.1%)
					Placebo	152	39 (25.7%)
Rubino DM, 2022 [20]	STEP-8	Randomized, placebo-controlled, phase 3b trial	Adults without diabetes and with a BMI of 30 kg/m^2^ or greater OR a BMI of 27 kg/m^2^ or greater with at least one comorbidity	68 weeks	Semaglutide 2.4 mg	126	24 (19%)
					Placebo	85	19 (22.4%)

## Data Availability

Data will be made available upon reasonable request.

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
