# Peer review of "The Effect of Semaglutide on Blood Pressure in Patients without Diabetes: A Systematic Review and Meta-Analysis"

_jcm, 2023, doi:10.3390/jcm12030772_

Round 1
Reviewer 1 Report
Important area of research. Meta-analysis summarizes data for effects of GLP-1 agonists on blood pressure in patients without diabetes. Obese patients without diabetes might benefit from this important off-target effect of GLP-1 agonists.
Well written manuscript. Search strategy for the meta-analysis was appropriately described. The authors addressed potential bias, performed sensitivity analysis to test the strength of their findings. The authors focus on the overall effects size, which compares well across studies.
A point of critique is the switching between fixed-effect and random effects model when data for systolic and diastolic blood pressure is summarised. Particularly in light of similar findings with both models. Schools of thought differ quite dramatically. Yet, as different study populations are included and generalisation to an even bigger population is aimed for, using a random effects model seems reasonable for this analysis. For comparison, the fixed-effect model may be published alongside in the supplemental material.
The authors should consider reducing the number of decimals to 2, unless p<0.01.
Author Response
Thank you for your positive comments. In response to your comments:
- The pooled analysis is now presented as the random effects model for both systolic and diastolic blood pressure. The fixed effects models are presented in the Supplementary Material
- Line 105, 107-110, 119-122, 153
- P values are now to two decimal places. Lines 159, 160, 162-3.
Reviewer 2 Report
The article: The Effect of Semaglutide on Blood Pressure in Patients With-2 out Diabetes: A Systematic Review and Meta-Analysis” is of interest since it provides new data about the reduction of blood pressure in obese non-diabetics treated with semaglutide. The authors have included six studies, detecting a 5-mmHg reduction in systolic BP and 2.5 mmHg for diastolic BP.
As the authors mention this study, together with the previous data, brings a new paradigm for the treatment of hypertension in patients with obesity.
The authors have used an adequate methodology for the systematic review (especially Covidence with Cochrane criteria) and the results obtained are compelling.
However, once the effect of semaglutide on blood pressure reduction has been demonstrated through the systematic review, it is necessary to make some considerations in the discussion:
Clearly, a distinction must be made between the O'Neil study (with low dose of semaglutide) and the studies of the STEP project (all with 2.4 mg of semaglutide), the latter with a greater weight due to a greater number of patients and a higher dose of semaglutide. That aspect is not sufficiently mentioned in the discussion. There is a clear greater effect in lowering blood pressure in patients in the STEP studies compared to the O'Neil study.
The limitations of the study have been clearly defined by the researchers, but they must also include that the effect of blood pressure has only been analyzed at the end of the study and there are no intermediate points during the study.
Finally, the authors make inferences with other studies where BP is reduced with GLP1 RA, but they do not clarify the possible mechanisms of blood pressure reduction, and if they are mediated solely by weight reduction or if there are other possible direct mechanisms on the kidney (anti-inflammatory, natriuretic, slimming, etc...) or other systems.
In the case of GLP 1-RA there are several studies that perform mediation analysis and demonstrate the weight of each of the factors (weight, blood pressure, direct renal effect, metabolic control) on the overall benefit (but in diabetics).
See the references below (in diabetics)
It could be interesting to mention in the discussion the possibility of carrying out a mediation analysis to explain the possible mechanisms of action in reducing blood pressure. As well as commenting on the natriuretic effect of GLP-1 RA, even if only initially.
https://dom-pubs.onlinelibrary.wiley.com/doi/10.1111/dom.14443
https://pubmed.ncbi.nlm.nih.gov/32366578/
https://pubmed.ncbi.nlm.nih.gov/33115821/
https://www.mdpi.com/1420-3049/26/16/4822
Please consider adding to the discussion more about the potential mechanisms of benefit in lowering blood pressure and how much benefit is related to weight loss.
Author Response
Thank you for your positive comments for this manuscript and the very helpful suggestions. In response:
- In the manuscript, the STEP trials are pooled and presented separately from the study by O’Neil in Figure 4 and in the Results (Lines 151-2). Excluding the latter study does not alter the effect on BP due to semaglutide. This result is then discussed in the manuscript (Lines 212-4).
- The limitation identified by the reviewer has now been included in the manuscript (Lines 302-305).
- Thank you for suggesting a very interesting addition to the manuscript. The possible mechanisms by which obesity and hypertension interact are included in the Introduction and contemporary opinion is referenced (Lines 39-43). Building on this further, we have added a paragraph on the examination of GLP1-RAs and their effect on BP, as well as the excellent suggestion that a mediation analysis should be performed (Lines 265-276).
Reviewer 3 Report
The novelty of this manuscript consist in approaching the blood pressure control from other perspective targeting an efficient management of arterial hypertension and obesity for non-diabetic patients using the semaglutide therapy. From this point of view, the manuscript has a high scientific potential and impact.
The text of the article is well structured being relevant in the field and the manner of the presentation of the whole work is easy to follow, clear and the language is scientifically appropriate. The references are adequate and the latest studies were taken in consideration.
The conclusions are systematic and concise and they open new scientific directions for further studies.
Author Response
Thank you for your positive comments